# The Risk of Colorectal Polyps after Weight Loss Therapy Versus Obesity: A Propensity-Matched Nationwide Cohort Study

**DOI:** 10.3390/cancers15194820

**Published:** 2023-09-30

**Authors:** Hisham Hussan, Eric McLaughlin, Chienwei Chiang, Joseph G. Marsano, David Lieberman

**Affiliations:** 1Division of Gastroenterology and Hepatology, Department of Internal Medicine, University of California Davis, Sacramento, CA 95616, USA; 2Department of Biomedical Informatics, College of Medicine, The Ohio State University, Columbus, OH 43210, USA; 3Division of Gastroenterology and Hepatology, Department of Internal Medicine, Oregon Health and Science University, Portland, OR 97239, USA

**Keywords:** bariatric surgery, gastric bypass, sleeve gastrectomy, colorectal cancer, risk factor

## Abstract

**Simple Summary:**

Obesity is a strong risk factor for the development of colorectal cancer, with a higher risk in men compared to women. Bariatric surgery is the most effective weight loss method. However, studies suggest a lower risk of colorectal cancer in women but not in men after bariatric surgery. In this study, we find that bariatric surgery mitigates the effect of obesity on the risk of colorectal polyps in both men and women. Future studies are needed to understand why men remain at higher risk of colorectal cancer despite a lower risk of polyps after bariatric surgery.

**Abstract:**

Background: A fundamental understanding of the impact of bariatric surgery (BRS) on mechanisms of colorectal carcinogenesis is limited. For instance, studies report a reduced risk of colorectal cancer in females but not in males after BRS. We examined whether this sex-specific difference existed at the earlier polyp development stage. Methods: This retrospective cohort study included 281,417 adults from the 2012–2020 MarketScan database. We compared polyps rates on colonoscopy in four groups: post- vs. pre-BRS (treatment) to post- vs. pre-severe obesity (SO) diagnosis (control). We focused our main analysis on a propensity-matched sample that yielded a balanced distribution of covariates in our four groups (*n* = 9680 adults, 21.9% males). We also adjusted for important covariates. Results: Metabolic syndrome parameters improved after bariatric surgery and worsened after severe obesity diagnosis (*p* < 0.05). The rate of polyps was 46.7% at a median of 0.5 years pre-BRS and 47.9% at a median of 0.6 years pre-SO diagnosis. The polyps rate was 45.4% at a median (range) of 3.2 (1.0–8.5) years post-BRS. Conversely, 53.8% of adults had polyps at 3.0 (1.0–8.6) years post-SO. There was no change in the risk of colorectal polyps in males or females post- vs. pre-BRS. However, the risk of polyps was higher in males (OR = 1.32, 95% CI: 1.02–1.70) and females (OR = 1.29, 95% CI: 1.13–1.47) post- vs. pre-SO. When compared to the control group (SO), the odds ratios for colorectal polyps were lower for males and females after bariatric surgery (OR = 0.63, 95% CI: 0.44–0.90, and OR = 0.79, 95% CI: 0.66–0.96, respectively). Conclusions: Obesity is associated with an increased risk of colorectal polyps, an effect that is ameliorated after bariatric surgery. These data are relevant for studies investigating colorectal carcinogenesis mechanisms.

## 1. Introduction

Obesity will soon surpass smoking and alcohol as a leading cause of preventable cancer in the United States and worldwide [1,2]. Obesity is a substantial risk factor for the development of colorectal cancer (CRC), the most frequent gastrointestinal cancer, affecting 2 million adults annually [3,4]. Notably, the risk of CRC is higher in males when compared to females with obesity [5,6]. Therefore, targeting obesity should help reduce the burden of CRC, especially in males. In that regard, bariatric surgery (BRS) offers the most impactful and sustained weight loss treatment for individuals with medically complicated obesity [7]. For instance, the weight loss conferred by Roux-en-Y gastric bypass is 21% higher at 10 years than non-surgical controls [8]. Metabolic parameters are also drastically improved after bariatric surgery [7,9]. Thus, through the lens of bariatric surgery, investigators will gain insight into the biology and impact of weight loss interventions on colorectal carcinogenesis.

Indeed, multiple studies investigated CRC after bariatric surgery. Yet, despite a suggested lower risk of CRC in females after bariatric surgery, prior epidemiological data do not confirm a reduction in CRC risk in males [10,11,12,13]. The lack of a response in males is despite a more pronounced weight loss than in females after bariatric surgery [14]. Understanding this biological effect of sex on the risk of colorectal neoplasia after bariatric surgery is critical for generalizability and informing future mechanistic and interventional studies investigating the interaction between sex, energy balance, and the risk of CRC. Notably, colorectal polyps can serve as early, intermediate surrogates of CRC risk. However, most prior studies examining the risk of colorectal polyps after bariatric surgery had a small sample size and did not assess if the risk of polyps varied by sex [10,15,16,17]. Therefore, we aimed to fill this knowledge gap by employing a propensity-score-matched, sex-stratified analysis using a nationwide database. We hypothesized that in adults with severe obesity, bariatric surgery is associated with a lower risk of colorectal polyps in both males and females when compared to no surgery.

## 2. Materials and Methods

### 2.1. The MarketScan Database

This was a retrospective cohort study using the IBM^®^ MarketScan Insurance Claims Research Databases, which has healthcare data for more than 39.7 million covered individuals and is one of the longest-running and largest collections of proprietary de-identified claims data for privately and publicly insured people in the U.S. [18]. Our study is based on automated claims data, which include confirmed age and sex. The MarketScan insurance claims are collected for reimbursement purposes. Therefore, we defined diagnoses and medications entered as billing claims as present. In contrast, in the absence of a diagnosis or medication insurance claim, we defined the disease or medication as absent. As a result, there were no missing data in our database, similar to other studies using MarketScan [19]. Data from MarketScan are de-identified and, thus, do not meet the federal definition of “human subject” per 45 Code of the Federal Regulation (CFR 46.101). Therefore, our study did not require review or approval by the Ohio State University Institutional Review Board.

### 2.2. Study Cohort

Details of our study design, inclusion, and exclusion criteria are in Figure 1. The 2012–2020 MarketScan database was queried using validated billing codes as published elsewhere [12]. First, we included adults ≥ 18 years of age with a colonoscopy and the diagnosis of severe obesity (SO). Severe obesity was defined as (1) a body mass index (BMI) ≥ 40 kg/m^2^ or (2) BMI ≥ 35 kg/m^2^ with obesity-related comorbidities. The above-mentioned definition of severe obesity is what is typically indicated for bariatric surgery and has been used in prior studies [11,20,21]. Then, we divided our cohort into a treatment and a control group. Our treatment group was defined as adults with SO who underwent either Roux-en-Y gastric bypass or vertical sleeve gastrectomy. Our control group was defined as adults with SO and no prior bariatric surgery. We further classified our treatment and control groups depending on the timing of colonoscopy, leading to 4 cross-sectional groups: adults with colonoscopy post-BRS or post-SO and adults with a baseline colonoscopy pre-BRS or pre-SO diagnosis [11,20]. We excluded patients with CRC risk factors other than severe obesity (e.g., personal history of CRC or polyps). We also excluded other less commonly used bariatric surgeries or gastric surgeries performed for reasons other than weight loss (e.g., gastric outlet obstruction or upper gastrointestinal ulcers). Finally, we did not include adults with colonoscopies performed within one year after BRS or SO to allow time for SO and BRS-induced weight change to take effect on the colon.

### 2.3. Outcomes

Our primary outcome was the difference in the rate of colorectal polyps post- vs. pre-BRS compared to post- vs. pre-SO. We investigated this outcome in the pre-matching and post-propensity-matching cohorts. A colonoscopy with a polyp was defined as a colonoscopy with polypectomy using cold forceps, snare, or mucosal resection, and an associated polyp diagnosis within 3 months after colonoscopy as previously defined [22]. A colonoscopy without polyp detection was defined as a complete colonoscopy without polypectomy and no polyps. In a sub-analysis, we assessed the risk of rectal polyps, which can be performed by restricting ICD codes to rectal polyps. Due to the absence of specific ICD-9-CM codes for colon polyps, it was not possible to specifically narrow the outcome to colon polyps or other anatomic locations.

### 2.4. Definition of Covariates

The Charlson comorbidity index (CCI) score, which accounted for number and severity of comorbidities, was used in our multivariable adjustment as was done in prior studies [10,11]. Alcohol or tobacco were defined by the presence of their respective billing codes at or prior to index visit. We also adjusted for colonoscopy screening versus diagnostic indications, which can confound the risk or detection of polyps on colonoscopy [23]. Type 2 diabetes and hyperlipidemia are components of metabolic syndrome, independently associated with an increased risk of CRC [24,25]. Therefore, we adjusted for the use of diabetes and cholesterol medications at or prior to the index visit, which can be done using MarketScan [26]. We defined being on diabetes or cholesterol medications as having at least two prescriptions belonging to these medications, at least 6 months apart, with one prescription date falling within 1 year prior to index visit date in the post-index date bariatric and control cohorts and prior to the pre-index colonoscopy in the pre-index cohorts [27].

### 2.5. Statistical Analysis

#### 2.5.1. The Pre-Matching Cohort

Our pre-matching cohort is described in Appendix A. Multivariable logistic regression was used to compare the odds of polyps post- vs. pre-BRS and post- vs. pre-SO in the unmatched cohort. Interactions were also utilized to compare post- vs. pre-BRS odds ratios to post- vs. pre-SO. Our multivariable model adjusted for age at colonoscopy, sex, Charlson comorbidity index, tobacco use, alcohol use, screening/non-screening colonoscopy, and colonoscopy performed < or ≥10/1/2015.

#### 2.5.2. The Propensity-Matched Analysis

Our matching flow is described in Appendix A. We propensity matched our four cross-sectional groups (post-BRS, post-SO, pre-BRS, and pre-SO) without replacement and using the greedy nearest neighbor methods to create a final analytical cohort matched 1:1:1:1. Propensity scores were calculated for each unique individual from the following variables: age at time of colonoscopy (within 2 years match, required), sex (exact match, required), Charlson comorbidity index individual components (diabetes without complications, diabetes with complications, myocardial infarction, congestive heart failure, peripheral vascular disease, cerebrovascular disease, dementia, chronic pulmonary disease, connective tissue disease-rheumatic disease, mild liver disease, paraplegia and hemiplegia, renal disease, cancer after excluding CRC and metastatic cancer, and moderate to severe liver disease), and colonoscopy performed < or ≥10/1/2015 (exact match, in order to account for potential coding bias related to transitioning from ICD-9CM to ICD-10CM codes) [28]. In the pre-BRS and pre-SO groups, length of time from colonoscopy to index was included in the propensity-matching between those groups, as were the times from index to colonoscopy in the post-BRS and post-SO groups. A caliper of 0.3 was used for matching, and standardized mean differences of the propensity score logit were assessed throughout the matching process to ensure adequate balance was generated. Chi-square and Kruskal–Wallis tests were reported to display the differences across groups in the cohort eligible prior to matching and the final cohort after matching was performed. Multivariable weighted logistic regression was used to compare the odds of polyps post- vs. pre-BRS and post- vs. pre-SO. Interactions were also utilized to compare post- vs. pre-BRS odds ratios to that post- vs. pre-SO. Our multivariable model adjusted for age at colonoscopy, sex, Charlson comorbidity index, tobacco use, alcohol use, screening/non-screening colonoscopy, colonoscopy performed < or ≥10/1/2015, and use of diabetes/cholesterol medications. Chi-square tests were used to compare diabetes and cholesterol medication use at index visit date and time of colonoscopy in the post-surgery patients and post-index controls. Multivariable logistic regression was also used to assess the odds of polyps in relation to diabetes medications. All statistical analyses were performed using SAS version 9.4 (SAS Institute, Inc., Cary, NC, USA).

## 3. Results

### 3.1. The Pre-Matching Population Analysis

#### 3.1.1. General Characteristics

Our pre-matching cohort included 281,417 adults (characteristics are described in Appendix A). In the pre-matched cohort, the SO adults were older than the BRS subjects (median of 55–56 vs. 52 years, respectively), with almost twice as many males and a slightly shorter follow-up post-index visit (median of 2.3 vs. 2.7 years, respectively); all statistically significant (*p* < 0.001). Tobacco and alcohol use trended towards higher frequencies pre-BRS and pre-SO, while screening colonoscopy indication trended to be higher in the post-index visit cohort (*p* < 0.001).

#### 3.1.2. The Risk of Polyps in Adults with or without Bariatric Surgery (the Pre-Matching Cohort)

In our 281,417 adults, the rates of polyps on colonoscopy were 47.3% pre-BRS and 51.8% pre-SO. At post-index colonoscopy, the rates of polyps were 45.9% post-BRS while 56.9% post-SO (polyp rates in males and females at pre- and post-colonoscopies are in Appendix A). Our unadjusted odds ratios are included in Appendix A. Our multivariable analysis (Table 1) showed reduced odds of colorectal polyps post- vs. pre-BRS (14% in males and 10% in females), while the odds of polyps increased post- vs. pre-SO (10% in males and 22% in females). As a result, the risk of colorectal polyps post-BRS was 22% lower in males and 27% lower in females than that of SO. However, in our sub-analysis, there was no change in the odds of rectal polyps pre- or post-SO or BRS, as in Appendix A.

### 3.2. The Matched Cohort Analysis

#### 3.2.1. General Characteristics

Our final propensity-matched cohort included 9680 adults (2420 in each cross-sectional group, Table 2). Our propensity-matched groups had the same age median (52 years) and male distribution (21.9%). The number of comorbidities was also clinically similar between our groups (median CCI score of 3). Comparable time elapsed between pre- and post-colonoscopy in the treatment and control groups. Differences in the unmatched covariates (screening colonoscopy indication, tobacco, and alcohol use) remained statistically significant between our matched groups, with a trend towards higher frequencies pre-BRS and pre-SO (*p* < 0.001).

#### 3.2.2. The Propensity-Matched Analysis Comparing the Risk of Colorectal Polyps in Adults with or without Bariatric Surgery

In our propensity-matched cohort of 9680 adults, the rate of colorectal polyps on colonoscopy was 46.7% at a median of 0.5 years pre-BRS and 47.9% at 0.6 years pre-SO (Figure 2). At the end of follow-up, the rate of polyps was 45.4% at a median (range) of 3.2 (1.0–8.5) years post-BRS. Conversely, 53.8% of adults had polyps at a median (range) of 3.0 (1.0–8.6) years post-SO. Our unadjusted odds ratios are included in Appendix A. After adjustment for all the variables in Table 2, date of colonoscopy, and also medication use at baseline, there was no change in the risk of colorectal polyps in males or females post- vs. pre-BRS (Table 3). Conversely, the risk of polyps was higher in SO controls after a similar follow-up period (OR = 1.32, 95% CI: 1.02–1.70 for males and OR = 1.29, 95% CI: 1.13–1.47 for females). As a result, the risk of colorectal polyps post-BRS was lower than that of severe obesity and no bariatric surgery (OR = 0.63, 95% CI: 0.44–0.90 for males and OR = 0.79, 95% CI: 0.66–0.96 for females).

#### 3.2.3. The Propensity-Matched Analysis Comparing the Risk of Rectal Polyps in Adults with or without Bariatric Surgery

Rectal polyps rates ranged between 7.9 and 8.9% in our propensity-matched cohort (Appendix A). When stratified by sex, the risk of rectal polyps was lower post- vs. pre-BRS in males but not in females as in Table 4 (OR = 0.66, 95% CI: 0.43–0.99 vs. OR = 1.11, 95% CI: 0.86–1.42, respectively). In controls with SO, there was no difference in the adjusted risk of rectal polyps during a similar follow-up. When compared to adults with bariatric surgery, there was a trend towards a reduction in rectal polyps in males, although that was not statistically significant.

### 3.3. The Impact of Bariatric Surgery on Metabolic Markers and Relation to Polyp Outcomes

We identify a reduction in diabetes and cholesterol medications’ usage at the time of post-BRS colonoscopy (e.g., a 24.3% reduction in males vs. 16.5% in females for diabetes medications as in Table 5, *p* < 0.001). In contrast, controls with SO were on more diabetes and cholesterol medications at the time of the post-SO colonoscopy (*p* < 0.001). Appendix A stratifies the rate of diabetes at index visits and changes in diabetes medications by sex post-bariatric surgery. When compared to our outcomes, we do not see an association between the odds of polyps and diabetes medications at index visits or at the time of colonoscopy, as in Table 6.

## 4. Discussion

This nationally representative, propensity-matched cohort study is the largest and first to evaluate sex-based differences in the risk of colorectal polyps after bariatric surgery as compared to persistent obesity. Per our propensity-matched analysis, the persistence of obesity was associated with an increased risk of polyps in both males and females—an expected result of worsening metabolic parameters. In contrast, the risk of colorectal polyps remained the same post- vs. pre-bariatric surgery, similar to our prior institutional data [29]. This finding is consistent with a recent meta-analysis showing an increased risk of CRC with weight gain but no conclusive change with weight loss [30]. Still, when compared to persistent obesity, bariatric surgery was associated with a lower risk of polyps, like a prior study that compared bariatric surgery to obesity controls [16]. Furthermore, our large sample size allowed us to stratify our sample by sex, where we noted a more pronounced effect in males who also had a lower risk of rectal polyps when comparing post- vs. pre-bariatric surgery in our propensity-matched analysis. We conclude that the persistence of obesity can increase the risk of colorectal polyps, an effect that can be ameliorated with bariatric surgery, especially in males, where the risk of rectal polyps is also suggestively reduced. These data strengthen the evidence for the deleterious effect of obesity on the risk of colorectal cancer in both males and females and the potentially protective effect of bariatric surgery.

In this study, we also investigated the change in diabetes and hyperlipidemia after bariatric surgery as compared to controls in both males and females. In this analysis, we used the change in the usage of diabetes medications to define new-onset diabetes (or its resolution) instead of disease codes since bariatric surgery can reduce weight and metabolic syndrome, making disease billing codes less precise after surgery. Our results show an improvement in diabetes and hyperlipidemia after bariatric surgery, while it worsened in controls with severe obesity. Nevertheless, we identified no relationship between metabolic improvement after bariatric surgery and the risk of polyps in males and females.

Our polyp data are consistent with accumulating evidence showing a lower risk of colorectal cancer in females undergoing bariatric surgery compared to controls with severe obesity [10,11,12,13]. Males also had lower odds of colorectal polyps compared to adults with persistent obesity. However, this finding in males does not conform with the epidemiological literature showing no change in CRC risk in males after bariatric surgery [13]. An explanation for a discrepancy between the risk of colorectal polyps and that of CRC in males after bariatric surgery compared to controls could be due to the lack of power or follow-up duration to detect a reduction of CRC in males. However, a recent meta-analysis with a supposably sufficient sample size shows a reduction in CRC in females but not in males followed for a similar amount of years [13]. Another theory is the beneficial effect of bariatric surgery on the colon but not rectosigmoid cancer [12]. This would be consistent with a known stronger effect of obesity on colon but not rectosigmoid cancer [31,32]. However, in this current study, there was a suggestive lower risk of rectal polyps in males post- vs. pre-bariatric surgery in our propensity-matched analysis, consistent with a more profound weight loss after bariatric surgery in males than females [33]. There was also an almost significantly lower risk of rectal polyps after bariatric surgery compared to severe obesity in males. This is, again, against data showing an almost increased risk of rectosigmoid cancer in males compared to controls [12]. Therefore, further data are needed to understand the impact of bariatric surgery on colon and rectosigmoid cancer risk in males.

One possible theory for a persistently increased risk of CRC in males compared to controls despite a lower prevalence of polyps may be due to increased acceleration of colorectal carcinogenesis due to colitis-associated carcinogenesis, a distinct pathway from adenoma-associated carcinogenesis [34,35,36,37]. Indeed, murine and human studies also report an increased colorectal inflammation after bariatric surgery for unknown mechanisms [38,39,40]. Furthermore, the diagnosis of de novo colitis after bariatric surgery has been reported [41,42,43]. These phenomena are likely due to a reduced colonic butyrate, a fiber fermentation product of colon bacteria that suppresses colonic NF-κB activity inflammatory cytokines and can modulate Wnt signaling, the most activated pathway in CRC [44,45,46,47,48,49,50,51,52,53,54,55,56,57,58,59,60,61,62]. Certainly, fiber intake drops by 45% post-bariatric surgery to 6–17 g/day—below the recommended intake of 25–30 g/day [63,64,65,66,67,68,69,70,71,72]. In parallel, studies observe a reduction in fecal butyrate levels and the abundance of butyrate-producing bacteria after bariatric surgery [73,74,75,76,77,78,79]. We suspect a lower butyrate would be more pronounced in males, who are reported to consume 20% less fiber per kcal/day than females after bariatric surgery [80]. As a result, an altered colonic milieu with lower butyrate could counteract the beneficial effect of weight loss on the colorectum. Data are limited; therefore, future mechanistic studies need to assess this theory and the impact of bariatric surgery on adenoma- and colitis-associated carcinogenesis.

Strengths and limitations: An ideal method to examine our hypothesis would be to prospectively measure the risk of neoplastic polyps on colonoscopy in males and females randomized to bariatric surgery vs. no surgery [81]. However, this design would be cumbersome and likely yield a small sample size to detect sex-based differences. As an alternative, our study used a propensity score, which allows for quasi-randomization depending on the status of bariatric surgery and the timing of colonoscopy. By comparing bariatric surgery to controls with severe obesity, we accounted for the effect of obesity or weight loss during a similar follow-up period on the risk of colorectal polyps. Our propensity-matched pre- and post-cohorts had similar characteristics except for the timing of colonoscopy, either pre- or post-BRS or SO. We also used thorough exclusions when choosing appropriate controls without bariatric surgery in order to minimize confounding and isolate the effect of obesity/bariatric surgery. As a result, our matched pre-SO cohort had a comparable rate of polyps to the pre-BRS cohort (47.9% vs. 46.7%, respectively, *p* = 0.42). Despite our comprehensive analysis, we report a few limitations due to the nature of the MarketScan database. For instance, we are limited by our retrospective design and use of billing codes despite using previously validated codes. Our cohort also remains a convenience sample, limited to patients with commercial insurance who received colonoscopies either pre- or post-surgery. Our stringent exclusions may also limit the generalizability of this study to adults who were excluded from our study. Also, we could not assess the pathology of polyps in MarketScan, which can be either precancerous (adenoma or serrated polyps) or completely benign (hyperplastic). Still, there is a strong correlation between adenoma detection rate (ADR) and polyp detection rate, which is used as a surrogate of ADR in some countries [22,82]. Furthermore, there is no association between hyperplastic polyps and adipokines/obesity, unlike adenomas, which makes hyperplastic polyps less likely to be altered after surgical weight loss to confound our findings [83,84,85,86]. Finally, we could not account for the exact BMI values or duration of obesity prior to surgery or prior to documentation of obesity in our controls. However, we adjusted for markers of metabolic syndrome that correlate with the duration and severity of obesity [87]. Future prospective studies also need to account for other risk factors such as diet, exercise, and race/ ethnicity, which we could not include using the MarketScan administrative database.

## 5. Conclusions

In summary, obesity is becoming the number one healthcare concern in the world. The study using a nationwide database is designed to provide descriptive data addressing the differential impact of bariatric surgery on the risk of colorectal polyps according to sex. This information sheds light on the natural history of the adenoma–carcinoma pathway of colorectal carcinogenesis with obesity and after bariatric surgery. These data also provide sex-based data for mechanistic studies and the design of future interventional studies using bariatric surgery. These studies are specifically needed to understand why men remain at higher risk of colorectal cancer despite a lower risk of polyps after bariatric surgery, which would be pivotal for public efforts aimed at reducing the risk of CRC.

## Figures and Tables

**Figure 1 cancers-15-04820-f001:**
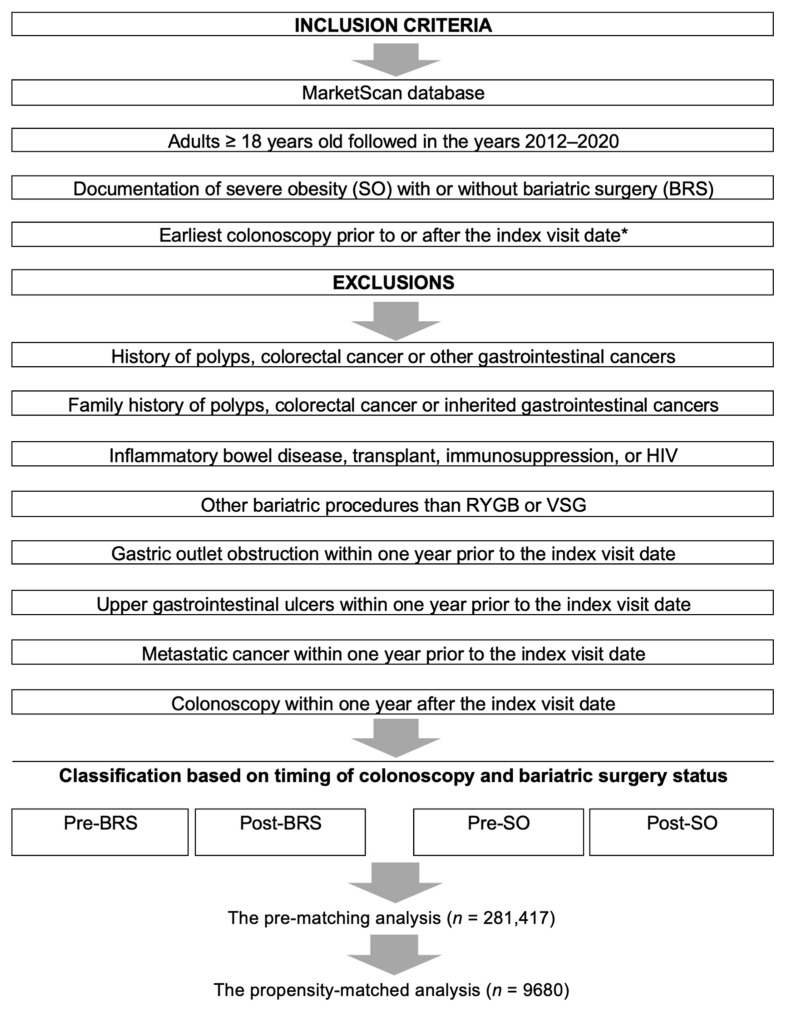
Study schema. * Index visit is earliest documentation of severe obesity for controls or date of bariatric surgery for cases.

**Figure 2 cancers-15-04820-f002:**
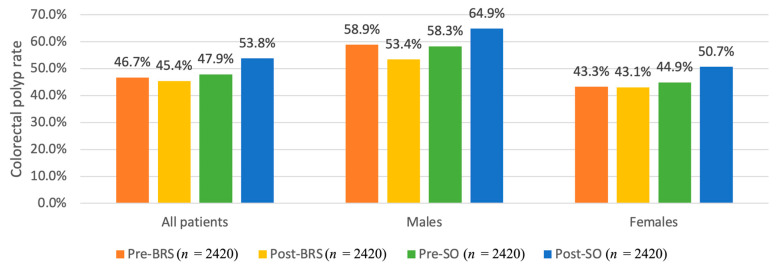
The propensity-matched analysis. Colorectal polyp rates by group and sex. BRS: Bariatric Surgery; SO: Severe Obesity.

**Table 1 cancers-15-04820-t001:** The full pre-matched cohort. Comparison of colorectal polyp odds ratios by sex pre- and post-bariatric surgery, pre- and post-severe obesity.

Post- vs. Pre- Colonoscopy		Males	Females
**Adjusted* OR** **(95% CI)**	BRS	0.86 (0.77–0.97)	0.90 (0.84–0.96)
	SO	1.10 (1.06–1.14)	1.22 (1.19–1.26)
	BRS vs. SO	0.78 (0.70–0.88)	0.73 (0.69–0.78)

* Models adjusted for age at colonoscopy, tobacco use, alcohol use, Charlson comorbidity index, screening colonoscopy, and date of colonoscopy (before/after 1 October 2015)

**Table 2 cancers-15-04820-t002:** Demographics of propensity-matched cohort.

Variable		Colonoscopy Pre-BRS (*n* = 2420)	Colonoscopy Post-BRS (*n* = 2420)	Colonoscopy Pre-SO (*n* = 2420)	Colonoscopy Post-SO (*n* = 2420)	*p*-Value
**Age at Time of Colonoscopy**		52 [49–57]	52 [49–57]	52 [49–57]	52 [49–57]	0.99
**Sex** **Male** **Female**		530 (21.9) 1890 (78.1)	530 (21.9) 1890 (78.1)	530 (21.9) 1890 (78.1)	530 (21.9) 1890 (78.1)	0.99
**Charlson Comorbidity Index**		3 [2–5]	3 [2–4]	3 [[2–4]	3 [1–4]	<0.001
**Years from Pre-BRS or Pre-SO Colonoscopy to Index Visit Date ***	**Median [IQR]**	0.5 [0.2–1.1]	N/A	0.6 [0.2–1.2]	N/A	0.003
	**Range**	0–6	N/A	0–6	N/A	
**Years from Index Visit Date to Post-BRS or Post-SO Colonoscopy**	**Median [IQR]**	N/A	3.2 [2.0–4.7]	N/A	3.0 [1.9–4.6]	0.19
	**Range**	N/A	1–8.5	N/A	1–8.6	
**Screening Colonoscopy** **Indication**		1475 (61.0)	1420 (58.7)	1380 (57.0)	1335 (55.2)	<0.001
**Alcohol Use**		39 (1.6)	11 (0.5)	34 (1.4)	23 (1.0)	<0.001
**Tobacco Use**		458 (18.9)	237 (9.8)	380 (15.7)	233 (9.6)	<0.001

Results are presented as count (column percentage) or median (first-third quartiles) BRS: Bariatric Surgery; SO: Severe Obesity. * Index visit date is earliest documentation of severe obesity for controls or date of bariatric surgery for cases.

**Table 3 cancers-15-04820-t003:** The propensity-matched analysis. Comparison of colorectal polyp odds ratios by sex pre- and post-bariatric surgery, pre- and post-severe obesity.

Post- vs. Pre- Colonoscopy		Males	Females
**Adjusted* OR** **(95% CI)**	BRS	0.83 (0.65–1.07)	1.02 (0.89–1.17)
	SO	1.32 (1.02–1.70)	1.29 (1.13–1.47)
	BRS vs. SO	0.63 (0.44–0.90)	0.79 (0.66–0.96)

BRS: Bariatric Surgery; SO: Severe Obesity. * Models adjusted for age at colonoscopy, tobacco use, alcohol use, Charlson comorbidity index, screening colonoscopy, date of colonoscopy (before/after 1 October 2015), diabetes medication use, and cholesterol medication use (medications in pre-index/BRS are at time of colonoscopy, medications in post-index/BRS are at time of index).

**Table 4 cancers-15-04820-t004:** The propensity-matched analysis. Comparison of rectal polyp odds ratios by sex pre- and post-bariatric surgery, pre- and post-severe obesity and in an interaction model.

Post- vs. Pre- Colonoscopy		Males	Females
**Adjusted* OR** **(95% CI)**	BRS	0.66 (0.43–0.99)	1.11 (0.86–1.42)
	SO	1.08 (0.72–1.61)	0.86 (0.68–1.09)
	BRS vs. SO	0.61 (0.34–1.08)	1.29 (0.92–1.81)

BRS: Bariatric Surgery; SO: Severe Obesity. * Models adjusted for age at colonoscopy, tobacco use, alcohol use, Charlson comorbidity index, screening colonoscopy, date of colonoscopy (before/after 1 October 2015), diabetes medication use, and cholesterol medication use (medications in pre-index/BRS are at time of colonoscopy, medications in post-index/BRS are at time of index).

**Table 5 cancers-15-04820-t005:** Analysis restricted to the propensity-matched post-index cohort. Medication usage from index visit to post-BRS or post-SO colonoscopy, stratified by sex.

Diabetes Medications’ Changes					
Group	N	Index Visit *	Post-BRS or Post-SO Colonoscopy	Difference	*p*-Value
**Male, BRS**	530	44.7%	20.4%	−24.3%	<0.001
**Male, SO**	530	24.2%	34.9%	+10.7%	<0.001
**Female, BRS**	1890	30.1%	13.6%	−16.5%	<0.001
**Female, SO**	1890	16.1%	28.2%	+12.1%	<0.001
**Cholesterol Medications’ Changes**					
**Group**	**N**	**Index Visit ***	**Post-BRS or Post-SO Colonoscopy**	**Difference**	***p*-Value**
**Male, BRS**	530	41.3%	29.3%	−12.0%	<0.001
**Male, SO**	530	26.6%	37.2%	+10.6%	<0.001
**Female, BRS**	1890	25.4%	17.6%	−7.8%	<0.001
**Female, SO**	1890	15.5%	25.0%	+9.5%	<0.001

BRS: Bariatric Surgery; SO: Severe Obesity. * Index visit date is earliest documentation of severe obesity for controls or date of bariatric surgery for cases

**Table 6 cancers-15-04820-t006:** Adjusted odds ratios for polyps post-BRIS stratified by gender in relation to diabetes mellitus *.

Variable	Post-BRS Males (*n* = 530)	Post-BRS Females (*n* = 1890)
**Diabetes medications at index** **(Reference = No)**	0.80 (0.54–1.17) *p* = 0.25	0.88 (0.71–1.10) *p* = 0.26
**Cessation of diabetes medications at post-index colonoscopy (Reference = Still use medication)**	0.83 (0.48–1.42) *p* = 0.50	0.91 (0.63–1.29) *p* = 0.59
**Began diabetes medication after index (Reference = No)**	1.04 (0.26–4.14) *p* = 0.96	1.00 (0.54–1.86) *p* = 0.99
**Diabetes medications at colonoscopy (Reference = No)**	1.04 (0.66–1.64) *p* = 0.86	0.98 (0.74–1.29) *p* = 0.86

* Models adjusted for age at colonoscopy, tobacco use, alcohol use, Charlson comorbidity index, screening colonoscopy, and date of colonoscopy (before/after 1 October 2015).

## Data Availability

Our detailed methods and codes are described in our paper. Patient-level, de-identified data were obtained from IBM MarketScan as part of a data agreement with the Ohio State University. Our investigators will make the analytical files available to any researchers for non-commercial purposes after the researcher obtains approval for third-party access from IBM MarketScan. Any researcher requesting access to the raw patient-level de-identified data that were used to generate the analytical files can access the data directly through IBM MarketScan under a license agreement with IBM MarketScan.

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
