# Peer review of "The Risk of Colorectal Polyps after Weight Loss Therapy Versus Obesity: A Propensity-Matched Nationwide Cohort Study"

_cancers, 2023, doi:10.3390/cancers15194820_

Round 1

Reviewer 1 Report

Thank you for the opportunity to review this work on risk of colorectal cancer pre and post bariatric surgery and pre and post sufficient weight loss.

My comments:

-In your abstract, just say propensity score matching yielded well-balanced distribution of covariates between patient cohorts. The sentences "Our groups..." to "...covariates" are unnecessary.

-Rather than doing propensity score matching, I would urge the authors to consider entropy balancing instead. This is a method that retains the entire cohort, whereas propensity score matching discards or matches observations which leads to lower sample size and reduced study power. Entropy balancing has also been shown to provide superior covariate balance compared to propensity score matching.

-If you choose to stick with propensity score matching, you should tell the readers whether it was 1:1 or 1:2, or something else, in addition to describing greedy nearest neighbor. Also tell the readers your caliper, so that replicability can be ensured.

-In table 1, you say your matched groups had similar characteristics, but you show no P value to substantiate this claim. I understand some covariates like age are nearly identical, but you have to show a P value. This is standard for any table 1 of any observational study.

-Consider showing a figure that shows pre and post propensity score matching covariate distribution. This is a helpful way to visualize the efficacy of your matching algorithm. If you decide to go with entropy balancing instead, a love plot would suffice.

-I do not see in the results any statement to the effect that we had N number of patients in the unmatched cohort which was reduced to X number of matched pairs following propensity matching. Readers need to be able to follow the flow of your propensity matching algorithm and understand the reduction in sample size following its application. 

Reviewer 2 Report

Authors investigated about bariatric surgery(BRS) on the risk of colorectal polyps according to sex, and in coclusion, described that ' these studies are specifically needed to understand why men remain at higher risk colorectal cancer despite a lower risk of polyps after bariatric surgery'.  The aim of the article was novel and worthy of an original article. However there were some questions as below to reach the conclusions. 

・Bariatric surgery was also called metabolic surgery due to its remarkable effect on the metabolic disorder especially on diabetes mellitus(DM), and DM was known as a strong factor of cancerization. However, authors didn't examine about DM of subjects ,its remission after BRS and its influence on cancerization. 

Therefore , authos were recommended to assess more in detail about the effect of bariatric surgery(BRS) on the subjects as below.

1 .  The ratio of DM  in subjects who underwent BRS in each gender respectively. 

2.   The effect of BRS on the DM of the subjects above .

3.   The relationship between DM or its remission after BRS and the polyp after BRS in each gender respectively. 

And then authors were recommended to discuss about the relationship among gender, DM and BRS. 

Reviewer 3 Report

Thank for the opportunity to review this manuscript. It is a well written paper about the risk of colorectal polyps after bariatric surgery compared to obese patients. The work has some limitations, such as the lack of histopathological diagnosis or follow-up, but they were listed and described in the discussion.

In my opinion the paper is worth publishing in Cancers. 

Reviewer 4 Report

The work as a whole is interesting.

Corrections must be made to the text.

In affiliation No. 1, it is removed in text 1. (The number 1 is repeated twice).

The bibliography in the text should be written better.

We recommend writing [1,2,3].

Insert the point after closing the parenthesis []. At present the point is before the parenthesis . [], ask for this to be fixed.

Line 154/161: The text has a different formatting from the rest of the article.

Table 2: p-value (no P-value). The "p" must be written small.

Round 2

Reviewer 1 Report

Sufficient revisions made

Reviewer 2 Report

Authors revised their article according to reviewer's comments appropriately.